# Dynamics of Cellulose Degradation by Soil Microorganisms from Two Contrasting Soil Types

**DOI:** 10.3390/microorganisms12081728

**Published:** 2024-08-21

**Authors:** Grigory V. Gladkov, Anastasiia K. Kimeklis, Olga V. Orlova, Tatiana O. Lisina, Arina A. Kichko, Alexander D. Bezlepsky, Evgeny E. Andronov

**Affiliations:** 1All-Russian Research Institute of Agricultural Microbiology, Podbel’skogo Highway 3, 196608 Saint Petersburg, Russia; akimeklis@arriam.ru (A.K.K.); falenki@hotmail.com (O.V.O.); lisina-to@yandex.ru (T.O.L.); 2014arki@gmail.com (A.A.K.); ad.bezlepsky@gmail.com (A.D.B.); eeandr@gmail.com (E.E.A.); 2Zoological Institute of Russian Academy of Sciences, 199034 Saint Petersburg, Russia; 3Department of Applied Ecology, Saint-Petersburg State University, 199034 Saint Petersburg, Russia; 4Institute of Physics and Mechanics, Peter the Great St. Petersburg Polytechnic University, 195251 Saint Petersburg, Russia; 5Dokuchaev Soil Science Institute, 119017 Moscow, Russia

**Keywords:** amplicon sequencing, chernozem soil, sod-podzolic soil, cellulolytic community, succession, cellulose decomposition

## Abstract

The search for active cellulolytic consortia among soil microorganisms is of significant applied interest, but the dynamics of the formation of such communities remain insufficiently studied. To gain insight into the formation of an active cellulolytic community, the experiment was designed to examine the colonization of a sterile substrate (cellulose) by microorganisms from two soil types: sod-podzolic and chernozem. To achieve this, the substrate was placed in the soil and incubated for six months. To assess microbiome dynamics, the experiment employed sequencing of 16S rRNA gene fragment and ITS2 amplicon libraries at four time points. It was demonstrated that, from the second month of the experiment, the prokaryotic component of the communities reached a state of stability, with a community composition specific to each soil type. The results demonstrated no relationship between changes in community diversity and soil respiration. There also was no significant shift in the community diversity throughout the chronosequence. Furthermore, the taxonomic composition of the community shifted towards a decrease in the proportion of Pseudomonadota and an increase in representatives of the Bacteroidota, Bacillota, and Verrucomicrobiota phyla. The network analysis of the community demonstrated that, in contrast to sod-podzolic soil, chernozem is distinguished by a higher modularity, with the formation of taxon-specific groups of microorganisms at each stage of the chronoseries. These differences are attributed to the alterations in the eukaryotic component of the community, particularly in the prevalence of nematodes and predatory fungi, which in turn influenced the cellulolytic community.

## 1. Introduction

The study of cellulolytic microbial communities is a highly popular topic due to its applied importance and the well-developed methodologies that enable the effective functional attribution and validation of results [1]. It should be noted, however, that a considerable number of these studies concentrate on either the rumen microbiome or the construction of particular strain combinations [2,3]. Despite not being a rare topic to be explored, there is still a lack of representation of soil microorganisms with cellulolytic activity in both research studies and databases (CAZy [4], PULDB [5,6]. Furthermore, most of these studies concentrate on the degradable substrate or the characterization of a particular cellulolytic ecotope [7]. The microbial cellulolytic component of the substrate and the microorganisms from the surrounding microbial pool cannot be separated in the communities thus obtained. This, in turn, prevents us from discerning the role of biotic and abiotic components, and understanding the interactions between different trophic groups within the community, and thus determines the sustainability and efficiency of the microbial community [8]. In this study, we employ a methodology that enables the investigation of the active soil microbiome through the analysis of the chronosequence involved in the colonization of a sterile substrate in nylon bags placed into soil. This approach represents a synthesis between traditional, culture-based techniques and standard procedures utilizing amplicon sequencing. In contrast to the simple mixing of soil with a substrate, our methodology reduces the background of dominant soil microorganisms, which are frequently not actively cellulolytic [9].

In a prior experiment examining community succession on a sterile substrate [10], soil microorganisms colonized oat straw, a material rich in simple sugars, proteins, and polysaccharides [11]. Simple sugars stimulate bacterial growth in the initial stages of colonization, and proteins provide nitrogen, which is often a limiting resource in the decomposition of lignocellulosic substrates [12]. In that study, a change in the prokaryotic community was observed in the middle phase of the substrate decomposition, during the transition from more accessible substrates (simple sugars) to polysaccharides.

Our study’s primary objective was to differentiate between the cellulolytic potential of two distinct and contrasting types of soil communities, which have not been previously accomplished. Sod-podzolic soil and chernozem are the most widespread soil types in Russia [13], but they differ contrastingly in chemical properties and development. The properties of their microbiomes are also contrastingly different [14]. While podzolic soils are typically found in undeveloped areas, chernozem is the main soil type used for plowing. The increase in cropland areas in Russia requires the development of precisely poor sod-podzolic soils, where the application of traditional practices is often not appropriate and the use of microbiological preparations becomes economically reasonable, e.g., when using no-till technology, where large amounts of organic residuals are left behind [15]. The utilization of crystalline cellulose as a substrate allowed for the investigation of an isolated community with a high catalytic potential for the development of highly efficient cellulolytic consortia based on it.

## 2. Materials and Methods

To set up the experiment, suspensions of sterile crystalline cellulose powder were placed in sterile nylon bags. Following this, the bags were placed at a depth of 5 mm in two types of soil: chernozem and sod-podzolic soil. The experiment was conducted at room temperature and a fixed humidity of 60% in 2 L plastic containers for six months. At 3, 28, 91, and 161 days, the bags were removed from the soil. Three replicates were used for each data point, with each replicate corresponding to a different bag. At each sampling point, soil respiration was quantified compared to a control sample comprising an identical soil mixture lacking cellulose. The methodology has been previously described in detail [16].

The soil DNA was isolated from the decomposed substrate, and the variable regions of the 16S rRNA gene and ITS2 region were sequenced on the Illumina MiSeq platform. The previously described routine methodology [17] was employed for the processing of sequencing. Furthermore, additional filtering was employed, whereby ASVs that had not been defined to the phylum level were removed from the dataset. To investigate the network organization of communities, covariance networks were constructed for the two soil types (sod-podzolic and chernozem). The networks demonstrate the co-occurrence of phylotypes for the chronoseries. Network analysis was conducted on the remaining ASVs, which were represented by more than five reads in no more than 10% of all samples. The SPIEC-EASI algorithm [18], as part of the SpiecEasi package within the R language, was employed for the construction of the networks. Meinshausen–Buhlmann’s neighborhood selection method was used [19]. The igraph package [20] was employed to calculate clusters and modularity of the network structure, while the random walk algorithm was utilized to select clusters. The GGally package was used for graph visualization, and normalization using the ANCOM-BC library was carried out to visualize the taxonomy and representation of groups within taxa [21]. The versions of the packages and the code used for post-processing can be found in the Appendix A and repository (https://github.com/crabron/dyn_cell, accessed on 15 July 2024).

## 3. Results

### 3.1. Soil Properties

The soils used for the experiment were contrastingly different in their properties (Appendix A). The sod-podzolic soil, due to its granulometric composition, belongs to the class of sandy loam soils (physical clay content, i.e., particles < 0.01 mm—16.6%), and chernozem belongs to the heavy loamy soils (physical clay content—52.88%). The granulometric composition of soil has a significant influence on soil formation and the agro-productive properties of soils. It determines the processes of the movement, transformation, and accumulation of substances as well as the physical, physical–mechanical, and water properties of soil, such as porosity, moisture capacity, water permeability, water holding capacity, structure, air, and thermal regime. These properties also affect the soil microbiome [22].

Podzolic soils are characterized by acidic and strongly acidic reactions (pH_sol_. 3.5–5.0), a low cation exchange capacity, a low saturation with bases (15–20%), and a low humus content (1–3% in the upper horizon) of the fulvate composition. In our case, the soil was cultivated with a high content of total organic carbon and a neutral pH value. However, the saturation of bases, in particular Ca^2+^ content, was much lower (almost 10 times lower) than in the chernozem. Chernozem was almost a benchmark of soil fertility, and this is evident from the data obtained: its pH was higher than 7, and the total organic carbon content was 8.75% (a fallow from the reserve was used as organic carbon is usually lower in cropland chernozems), and it contains a lot of calcium. The content of mobile forms of phosphorus and potassium was also higher in the chernozem.

### 3.2. Amplicon Sequencing

Two sets of amplicon libraries were obtained—for the variable region of the 16S rRNA gene (78,400 reads) and ITS2 fragment (117,110 reads). The most represented phyla in the prokaryotic community were Pseudomonadota, Bacteroidota, Bacillota, Verrucomicrobiota, and Actinobacteriota. Chronoseries were similar at the phylum level for the different soil types. The early stages were characterized by representatives of Gammaproteobacteria, Bacillota, and Actinobacteriota. In the late stages, Alphaproteobacteria, Bacteroidota, Verrucomicrobiota, Planctomycetota, and Myxococcota were present.

The increase in the alpha-diversity indices was a characteristic feature of sod-podzolic soil, whereas a decline was observed in the chernozem (Figure 1). While the alpha-diversity indices for different soil types demonstrated opposing dynamics, the respiration profile exhibited a consistent pattern, displaying a growth phase up to the third point and a subsequent slight decline. The lack of significant shifts in respiration and diversity dynamics can be attributed to the absence of a change in the community’s substrate preference, from simple sugars to complex substrates. Conversely, there is no evidence to suggest that the increase in respiration at day 91 is linked to the taxonomic shift in the community.

The resulting communities exhibited significant differences from one another (adonis *p*-value < 0.001) (Figure 2). The development of the prokaryotic and eukaryotic communities was strictly linear from the early to the late stages. The partitioning of the community by the substrate and correspondence between the eukaryotic and prokaryotic successions were observed. The data demonstrate that alterations in community dynamics occurred up to day 28, after which the taxonomic composition of the community reached a state of stability.

Community differences manifested at the high taxonomic levels. Bacillota, Chloroflexota, and Bdellovibrionota were more characteristic of the sod-podzolic soil. Chernozem was distinguished by the initial presence of Archaea and a considerably higher proportion of *Streptomyces* relative to other organisms. Similar microorganisms were identified in the two distinct soil types, although their quantities varied throughout the stages. This phenomenon is particularly evident among Acidobacteriota. Concurrently, at the low taxonomic level (ASV, genus), the soil samples exhibited notable differences, particularly among representatives of the Bacteroidota phylum. This observation can be attributed to the high functional diversity observed among members of this phylum [23].

The identified eukaryotic community displayed a considerably lower level of diversity than the prokaryotic community (334 vs. 2427 ASVs), but its dynamics exhibited correspondence with the prokaryotic community. In contrast to the sod-podzolic soil, the data for chernozem indicate differences at day 28 from later phases. At day 28, the community included basidiomycetes *Coprinellus*, ascomycetes *Chaetomium* and *Parachaetomium* (previously identified as cellulolytic [24], and *Schizothecium* and *Dictyosporium*. In the late stages, a high proportion of nematodes was observed in the community, while the main fungus was represented by a predatory fungi from the genus *Arthrobotrys* [25]. The cellulolytic component of the community was represented by members of the genus *Stachybotrys*, which emerged at the second point of the chronoseries [26].

In both soil types, by the late stages, there was an increase in the relative representation of nematodes in comparison to the remainder of the eukaryotic community. In the sod-podzolic soil, the proportion of nematodes among the total number of eukaryotes increased from the second to the third stage, from 29.5 to 41.5 percent. In the chernozem soil, the same trend was observed, but with a lower range, from 8.1 to 16.2 percent. Concurrently, nematodes from disparate ecological niches were represented in the two soil types. For example, in the sod-podzolic soil, *Acrobeloides*, *Pseudacrobeles*, and *Aphelenchoides* [27] were present. For the sod-podzolic soil, the nematodes observed were *Acrobeloides*, *Pseudacrobeles*, *Aphelenchoides* (only in the second stage), *Ditylenchus medicaginis*, and *Pseudacrobeles curvatus* (only in the third stage). For the chernozem soil, the nematodes observed were *Acrobeloides*, *Pseudacrobeles*, *Aphelenchoides* (only in the second stage), *Ditylenchus medicaginis*, and *Pseudacrobeles curvatus* (only in the third stage) [26].

### 3.3. Co-Occurence Networks

We applied a compositional approach to construct co-correlation networks (Figure 3), which enabled us to normalize the varying amplicon library sizes while avoiding the distortions associated with rarefaction or normalization. It was demonstrated that the resulting network structure varied according to the soil type. The communities obtained from the sod-podzolic soil displayed a greater structural homogeneity compared to those from the chernozem. The chernozem community exhibited a lower modularity and network connectivity (network modularity index for sod-podzolic soil: 0.28; for chernozem: 0.32). Clusters identified using a random walk algorithm showed analogous alterations in relative representation. In the case of sod-podzolic soil, eight groups of microorganisms were identified, four of which were classified as major (i.e., comprising more than 5% of the total community) (Figure 4). In the case of the chernozem soil, nine groups were identified, of which four were classified as major (Figure 5). The taxonomic composition of the sod-podzolic soil was relatively homogeneous, in contrast to the chernozem, which was characterized by the presence of small taxon-specific clusters. Cluster 6, which is characterized by a high prevalence of major ASVs belonging to the Bacteroidota phylum (specifically *Chitinofaga*, *Ohtaekwangia*, *Niastella*, and *Dyadobacter*), is worthy of closer examination. The most abundant bacteria in this cluster were specifically associated with the second point in the chronoseries, which we believe is directly associated with the main differences between the two soil types. Additionally, chernozem cluster 4, dominated by actinobacteria (*Streptomyces*), is also noteworthy, with representatives being the most abundant at the first point in the chronoseries.

## 4. Discussion

The application of metagenomic methodologies in the search for effective strains of biodecomposers is a common practice in the scientific community. However, the utilization of the obtained strains in biopreparations frequently demonstrates considerable instability of active microorganisms when applied to soil. An alternative approach involves the use of complex biopreparations [28,29]. This strategy requires a comprehensive understanding of both the decomposition processes and the ecological impacts specific to the soil type to which the biopreparation is applied [30]. This study focuses on the temporal dynamics of sterile crystalline cellulose decomposition, with a particular focus on the ecological implications of cellulolytic activity among soil microorganisms, particularly concerning the soil type in which the substrate is applied.

Previously, we conducted a similar study using a more biologically accessible substrate (oat straw) in a single soil type (sod-podzolic). The results demonstrated a clear correlation between the shift in the prokaryotic community composition from simpler substrates (simple sugars) to more complex ones (polysaccharides) and the change in the total soil respiration. In this study, cellulose was selected as the substrate to eliminate the influence of the source material and to focus on the role of soil microorganisms in the degradation of complex polysaccharides. The chronoseries on crystalline cellulose demonstrated a relatively stable diversity over time. Additionally, the inverse correlation between alpha-diversity indices and respiration, previously observed in other studies [10], was not evident in this experiment. The literature shows that the presence of bacterivorous nematodes in the soil leads to an increase in soil respiration [31], but we were unable to detect this effect.

In contrast to the minimal alteration in the alpha diversity indices, the prokaryotic community structure underwent a significant transformation throughout the study period. Our findings demonstrate the presence of cellulolytic microorganisms in both the early and late stages. Early on, we identified many of the cellulolytic microorganisms described in the literature, such as representatives of the genera *Streptomyces* and *Paenibacillus* for chernozem, and *Ochtaekwangia* for sod-podzolic soil [32]. By the later stages, the proportion of Pseudomonadota had decreased, while the proportion of minor Bacillota representatives and the total proportion of Bacteroidota and minor phyla had increased.

These observations are largely aligned with the previous reports. For instance, the initial colonization of the site by members of the *Burkholderia*, *Pseudomonas*, and *Streptomyces* genera [33] has been previously documented. Notably, a subset of Bacteroidota (*Niastella*) was present in the initial stages but was subsequently displaced from the community. For each substrate species, two distinct stages of microbial community development can be identified: an early stage (3 days) and a late stage (28–161 days). In the case of the eukaryotic amplicon libraries of chernozem, an intermediate stage at day 28 is also distinguished, which is absent in the sod-podzolic soil community. At this stage, a decline in soil respiration, a reduction in the diversity of prokaryotes, and a shift in the taxonomic profile of eukaryotes were observed.

It has been shown that it is possible to use a network approach to community analysis to investigate these types of interactions, due to the weakness of the observed effects (especially at high taxonomic levels such as phyla, and families). The changes in the prokaryotic profile, as revealed by the network analysis, appear to be indirectly associated with modifications in the eukaryotic component. It is hypothesized that the key drivers of the community formation were the presence of specific nematodes in the soil and their subsequent interactions with fungal and prokaryotic components. Despite the relatively low diversity (only 11 phylotypes), nematodes were found to account for 41.5% of all the eukaryotic reads in the sod-podzolic soil. The predominant group of reads belonged to the predatory fungus *Arthrobotrys*, which prey on nematodes. This fungus was present in the sod-podzolic soil in the initial stage and in the chernozem soil in the third stage, with a representation of 64% in the latter. Notably, the predominant nematodes in different soil types belong to disparate trophic groups. For example, *Acrobeloides* and *Pseudacrobeles*, prevalent in sod-podzolic soil, are bacteria feeders [27], whereas *Ditylenchus* and *Aphelenchoides* are mycotrophs or plant parasites [34]. A specific response of the prokaryotic component to the overrepresentation of nematodes in the second stage of chernozem has been demonstrated. The presence of *Vampiriovibrio* was observed in the second stage for the chernozem, in contrast to the sod-podzolic soil, where *Chitinofaga* was the predominant species. These microorganisms have been associated with the decomposition of fungal mycelium residues [35] and predation [36].

The observed differences in the substrate occupation by soil cellulolytic organisms between the two soil types can be linked to the fact that the sod-podzolic soil is distinguished by a higher prevalence of slow-growing microorganisms with high catalytic potential, in contrast to the more nutrient-rich chernozem [37]. Additionally, the dissimilarity between the two soil types can be explained by the fact that the chernozem was a less diverse source of microorganisms (as indicated by lower alpha diversity indices of the original soil [38], which resulted in the longer formation of a stable community. These findings are consistent with the previously described phenomenon, wherein the microbial community’s response to a eukaryotic component serves as a site-specific marker [39]. Some studies indicate that this response is closely correlated with the available soil carbon levels [40], and there is evidence that the main predictor of nematode growth is the presence of macroaggregates in the soil [41], which in turn already stimulates nitrogen metabolism in the soil. Since sod-podzolic soil and chernozem differ in particle size, we most likely observed a response to this phenomenon in the dynamics of the microbial community.

## 5. Conclusions

This study demonstrates the distinction between primary colonization by soil cellulolytic organisms originating from two contrasting soil types: chernozem and sod-podzolic soil. The community structure of microorganisms colonizing the sod-podzolic soil substrate exhibited a higher degree of taxonomic homogeneity. Additionally, a gradual increase in the specificity of the prokaryotic component of the communities was observed, contingent upon the dynamics of the eukaryotic component.

## Figures and Tables

**Figure 1 microorganisms-12-01728-f001:**
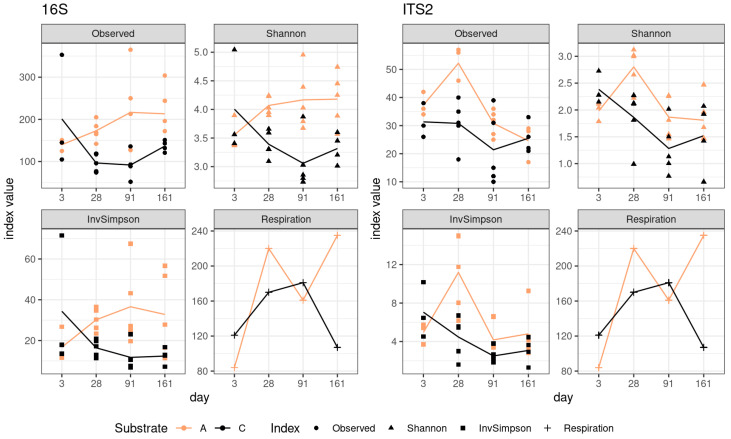
Variation in alpha diversity and respiration indices. A (light)—sod-podzolic soil. C (dark)—chernozem.

**Figure 2 microorganisms-12-01728-f002:**
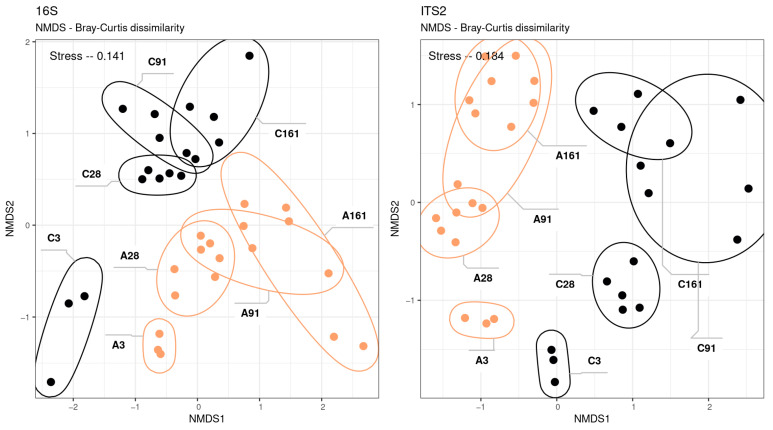
NMDS—beta diversity; A (light)—sod-podzolic soil; C (dark)—chernozem: left for 16S rRNA gene and right for ITS2 fragment libraries.

**Figure 3 microorganisms-12-01728-f003:**
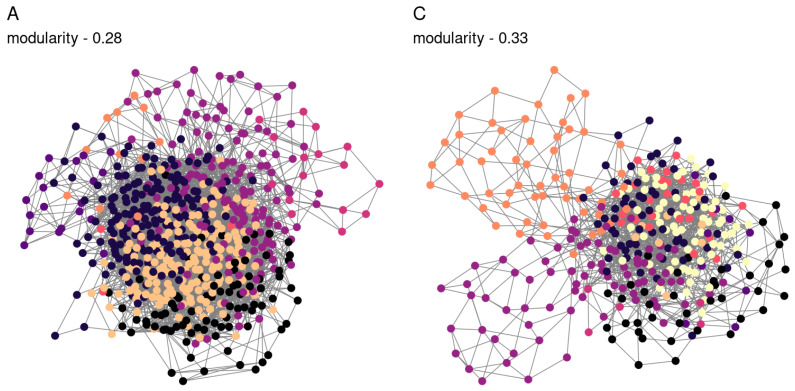
Co-correlation networks. Color—selected modules of co-correlation of abundance change. A—sod-podzolic soil; C—chernozem. The network was obtained using SPIEC-EASI algorithm, and modules were selected using the random walk algorithm.

**Figure 4 microorganisms-12-01728-f004:**
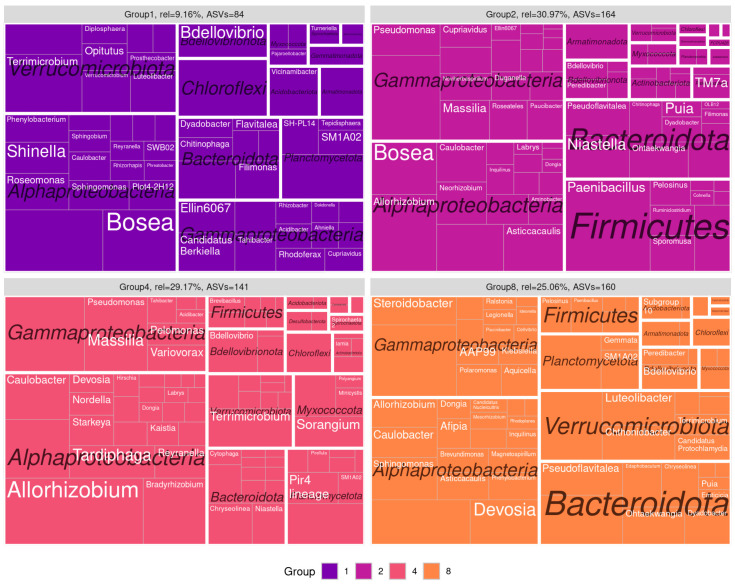
Taxonomic composition of the clusters of the co-correlation network for sod-podzolic soil. Clusters with relative representation of more than 9% are visualized. Taxonomic levels represented are phylum and genus. The tile size reflects the relative representation of ASVs within the genus.

**Figure 5 microorganisms-12-01728-f005:**
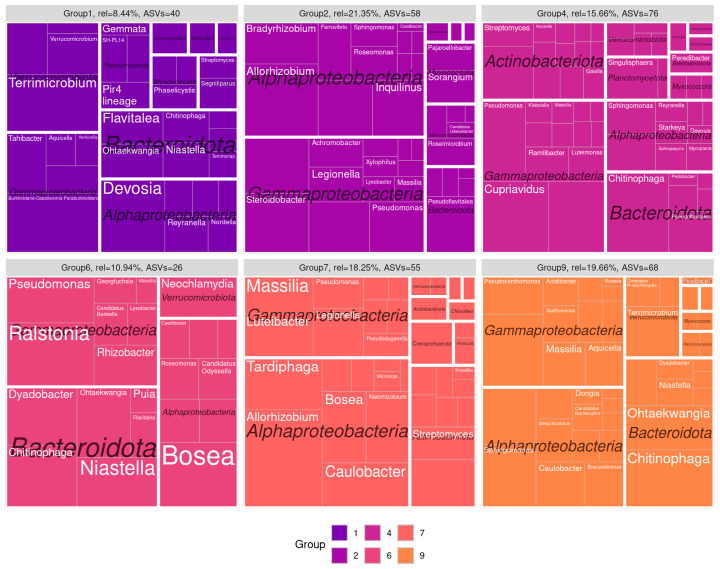
Taxonomic composition of the clusters of the co-correlation network for chernozem. Clusters with relative representation of more than 8% are visualized. The represented taxonomic levels are phylum and genus.

## Data Availability

Data are available under the NCBI BioProject ID number PRJNA841641.

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
