# Peer review of "Dynamics of Cellulose Degradation by Soil Microorganisms from Two Contrasting Soil Types"

_microorganisms, 2024, doi:10.3390/microorganisms12081728_

Round 1

Reviewer 1 Report

Comments and Suggestions for Authors

The manuscript describes a metagenomic approach to determine the structure and diversity of the microbial community that colonizes crystalline cellulose substrate in two soil types, sod-podzolic and chernozem. For metagenome sequensing, soil samples were taken at the 7, 35, 70 and 98 days, in the initial times microbial community diversity was relatively stable, main changes were observed after day 35, mainly in prokaryotes. The community's taxonomic composition shifted towards a decrease in the proportion of Proteobacteria and an increase in representatives of the Bacteroidota, Bacillota, and Verrucomicrobiota phyla. The network analysis of the community demonstrated that, in contrast to sod-podzolic soil, chernozem is distinguished by a higher modularity, with the formation of taxon-specific groups of microorganisms at each stage of the chronoseries.

The authors must address the following commentaries:

A better description of the rationale for selection of the soil employed for experiments is needed, which are their characteristics (mineral composition, pH, moisture, organic matter proportion, presence of other nutrients such N and P), why it is important characterize cellulolytic communities in these soil, which advantages bring respect applicability. In the geographic context of the study, are these soils representative?

Explain in discussion and highlight in conclusions how the research results help to stablish better cellulose degradation processes, how the generated knowledge has applicability, or how it can generate cellulolytic consortia, as was proposed at the end of the manuscript's introduction.

Author Response

1) The authors developed a study based on ASV species paradigm, which requires the construction of error models for denoising of sequences to the ASVs of the system. The authors worked with DADA2, which by default uses the first 100000000 bp from alphabetically-ordered samples to construct this model. Since the authors did not mention anything about this, I will assume they used that amount of bp to develop the DADA2 error model. How many sequences were used for error model construction? What percentage is that with respect to the total number of sequences analyzed? It is logical that inaccurate error models constructed with a very low number of sequences will lead to inaccuracies in the microbial ecology analysis conducted.

Thanks for your question, the model construction with dada2 is the bottleneck for analysis. For the default fourth part of data (our dataset is not so big) used for model construction, we used full data and the results seem the same (review Supplement, "Error model" part).

2) What are the parameters for quality filtering? Were ambiguous bases permitted? What about homopolymeric sequences? What was the minimum overlap allowed for merging of paired-end reads? This data is required to provide replicable science.

"What about homopolymeric sequences?" - Thanks for the amazing question. As known, for ILLUMINAs' data this kind of error is rare, but we tried an additional (but also kind of controversial) check based on the data that we added in an additional file that you can read in the revision supplement in the "Homopolymeric Sequences" part.

After the next question block are the answers to the main questions.

3) What algorithms were used for chimera detection and phylogenetic classification? Please remind that saying "done using DADA2" is not declaring an algorithm but a software. It would be the same as saying you developed a computation using Windows rather than your mathematical software of reference.

The script that is used for the DADA2 pipeline is in the github repository mentioned in the methods, all the R code used for the post-processing is in Supplement 1 and the raw code is also in the repository.  The version of the system and packages used in the analysis are also included in the rendered rmd file of the supplement.

4) Most importantly, the statistical approach is all wrong because no compositional statistics were used. Microbiome data is compositional and analyzing it without considering this mathematical property is inherently wrong and makes the study prone to errors when studying everything the authors did except for alpha-diversity. If the authors would like to conduct a proper statistical work with the microbiome data they need to use compositional statistics.

There is a possibility of misunderstanding. This is because we used a compositional approach to build the network. The SPIEC-EASY pipeline is based on a compositional statistical approach. We also tried some compositional (e.g. CLR transformation) and related methods (e.g. ANCOMBC), but we didn't get any additional results (we tried WGCNA with compositional transformations for network construction before using SPIEC-EASY, but our data didn't have properties for such an approach), so we think it's too much for these data. Therefore, we left compositional analysis for network construction only where it was really needed.

For alpha diversity we used only rarefaction, but before I used https://github.com/adw96/breakaway based on the composition approach and it didn't change the results (see appendix suspicious_asvs.pdf for it). Also, we work with soil with big richness, so compositional effects are quite small, and pseudocaunts' misrepresentations change results dramatically (see https://doi.org/10.1128/msphere.00354-23). So, before network construction, we delete minors with quite a strict threshold, which is quite wrong and painfully compromised based on the result above.

Reviewer 2 Report

Comments and Suggestions for Authors

The authors based their research on environmental microbial analyses of soil communities.

However, according to the materials and methods presented by the authors, the microbial ecology analyses conducted appear to have so many flaws that it is impossible to accept this manuscript.

1) The authors developed a study based on ASV species paradigm, which requires the construction of error models for denoising of sequences to the ASVs of the system. The authors worked with DADA2, which by default uses the first 100000000 bp from alphabetically-ordered samples to construct this model. Since the authors did not mention anything about this, I will assume they used that amount of bp to develop the DADA2 error model. How many sequences were used for error model construction? What percentage is that with respect to the total number of sequences analyzed? It is logical that inaccurate error models constructed with a very low number of sequences will lead to inaccuracies in the microbial ecology analysis conducted.

2) What are the parameters for quality filtering? Were ambiguous bases permitted? What about homopolymeric sequences? What was the minimum overlap allowed for merging of paired-end reads? This data is required to provide replicable science.

3) What algorithms were used for chimera detection and phylogenetic classification? Please remind that saying "done using DADA2" is not declaring an algorithm but a software. It would be the same as saying you developed a computation using Windows rather than your mathematical software of reference.

4) Most importantly, the statistical approach is all wrong because no compositional statistics were used. Microbiome data is compositional and analyzing it without considering this mathematical property is inherently wrong and makes the study prone to errors when studying everything the authors did except for alpha-diversity. If the authors would like to conduct a proper statistical work with the microbiome data they need to use compositional statistics.

Without these amendments there is no possibility for this research to be published.

Author Response

A better description of the rationale for selection of the soil employed for experiments is needed, which are their characteristics (mineral composition, pH, moisture, organic matter proportion, presence of other nutrients such N and P), why it is important characterize cellulolytic communities in these soil, which advantages bring respect applicability. In the geographic context of the study, are these soils representative?

Explain in discussion and highlight in conclusions how the research results help to stablish better cellulose degradation processes, how the generated knowledge has applicability, or how it can generate cellulolytic consortia, as was proposed at the end of the manuscript's introduction.

We added a section in the results describing these soils. We also added additional reasoning to the introduction of why these soils were chosen. We modified the discussion slightly to respond to your last comment.

It should be noted that we have already tried to create effective soil communities based on these data (https://doi.org/10.3390/ijms231810779), but since the publication of the data does not correspond to the actual years of work (the published data have been lying unprocessed for quite a long time) there is a conceptual gap. I think this article has enough self-citations, I would not like to add one more.